# Effects of Glucose Oxidase and *Macleaya cordata* Extract on Immune Function, Antioxidant Capacity, and Gut Microbiota in British Shorthair Cats

**DOI:** 10.3390/metabo15120759

**Published:** 2025-11-24

**Authors:** Lizhen Li, Xuanzhen He, Tao Kuang, Zhuoting Chen, Yan Guo, Zhiyi Huang, Shiyan Jian, Zipeng Jiang, Limeng Zhang, Baichuan Deng, Qingshen Liu

**Affiliations:** 1Guangdong Provincial Key Laboratory of Animal Nutrition Control, College of Animal Science, South China Agricultural University, Guangzhou 510642, China; 10117llz@stu.scau.edu.cn (L.L.); hyunchun.ho@gmail.com (X.H.); kuangtao0904@163.com (T.K.); 13319608075@163.com (Z.C.); gy@scau.edu.cn (Y.G.); jianshiyan1029@163.com (S.J.); 2Guangdong VTR Biotech Co., Ltd., Zhuhai 519000, China; cattle33@163.com (Z.H.); 12117004@zju.edu.cn (Z.J.); 3School of Biology and Biological Engineering, South China University of Technology, Guangzhou 510642, China; 4Guangzhou Qingke Biotechnology Co., Ltd., Guangzhou 510080, China; 19865040001@163.com

**Keywords:** cat, glucose oxidase, *Macleaya cordata* extract, gut microbiota, pet food

## Abstract

Objectives: The objective of the present study was to investigate the effects of Glucose oxidase (GOx) and *Macleaya cordata* extract (MCE) on immune response, antioxidant capacity, gut microbiota, and metabolome in cats. Methods: Twenty-four cats were randomly divided into four groups: basal diet (CON group), basal diet + 0.03% GOx (GOD group), basal diet + 0.03% MCE (MCE group), and basal diet + 0.03% GOx and 0.03% MCE (GM group). Results: Compared to the CON group, the GOD group exhibited elevated levels of total antioxidant capacity (T-AOC) and secretory immunoglobulin A (sIgA), and decreased levels of interleukin-6 (IL-6) and immunoglobulin A (IgA) (*p* < 0.05). MCE increased concentrations of IgA, immunoglobulin G (IgG) and sIgA, alongside a reduction in interleukin-2 (IL-2). The GM group exhibited markedly elevated concentrations of IL-2 and IgG, and decreased levels of interleukin-10 (IL-10). Moreover, 16S rRNA sequencing showed differences in the fecal microbiota among the four dietary groups. Analyses of fecal and serum metabolomics demonstrated that differential metabolites were primarily associated with cat amino acid metabolism and fatty acid metabolism. Conclusions: These findings suggest that Gox and MCE may enhance immune function, mitigate oxidative stress in cats, and increase the relative abundance of beneficial gut microbiota. Moreover, our results may provide evidence for GOx and MCE as novel nutritional additives in pet food. It should be noted that this study is limited by its sample size; while the results provide promising insights, future studies with larger-scale studies are warranted to confirm these observations.

## 1. Introduction

Feed additives provide nutritional benefits, specifically supplementing key nutrients required by cats while regulating the balance of intestinal microbiota, thereby improving digestive absorption and maintaining intestinal health [1,2,3]. The population of domestic cats and dogs has seen a steady increase in recent years. Nowadays, cats and dogs are viewed more as family members rather than house guards. Pet owners are prioritizing high-quality nutritional care for their pets and are open to offering a range of services for their cherished companions [4]. Recently, the utilization of feed enzymes and plant extracts has become popular eco-friendly feed additives in animal nutrition [5,6].

Glucose oxidase (GOx), also known as β-D-glucose oxidoreductase, is an aerobic dehydrogenase that specifically catalyzes the conversion of β-D-glucose into gluconic acid while generating hydrogen peroxide [7]. GOx is primarily sourced from bacteria, fungi, and certain animals [], and it possesses various bioactive properties, including promoting growth in animals [8], improving intestinal environment through regulating the intestinal flora [9], reducing intestinal mycotoxicosis [10], and boosting antioxidant capacity and immunity [11]. Recent research indicates that dietary supplementation of GOx may enhance intestinal morphology and reduce apoptosis in intestinal epithelial cells [12,13].

*Macleaya cordata* is a perennial herbaceous plant classified within the genus Macleaya and the family Papavereraceae [14]. The extract from *Macleaya cordata* (MCE) is a phytogenic additive [15]. MCE has been used in various species due to its numerous biological effects, primarily attributed to its key active compounds, sanguinarine and chelidonine [16,17]. These alkaloids exhibit a variety of biological activities that have been verified in various mammals. They can selectively inhibit pathogenic bacteria and enhance intestinal barrier integrity by up-regulating tight junction proteins, while alleviating oxidative stress by activating the Nrf2 pathway. These effects collectively improve intestinal health, promote nutrient absorption and utilization, and exhibit antimicrobial, anti-inflammatory, and enzyme-modulating properties [18,19,20]. Empirical studies have shown that adding sanguinarine or chelidonine into the diet can enhance gut structure and antioxidant levels in both piglets and mice [21,22,23]. Additionally, there is evidence that MCE can enhance growth performance and immune function, and regulate the composition of gut microbiota in broiler chickens [24]. Other plant-derived bioactive compounds (such as Litsea cubeba essential oil, Astragalus polysaccharide, and Poria cocos polysaccharide) have shown similar results in related studies [25,26].

Although the impacts of GOx and MCE have been thoroughly researched in chickens and pigs, their application in cats—a species with distinct digestive physiology and nutritional requirements—is still unclear. Thus, this study aimed to test the hypothesis that dietary supplementation with GOx and MCE enhances immunity, antioxidant capacity, and beneficial shifts in fecal microbiota and metabolome profiles in cats.

## 2. Materials and Methods

### 2.1. Animals Housing and Breeding

All procedures were approved by the Animal Care and Use Committee before conducting the animal study (Approval number: 2021A030), in accordance with the guidelines established by the Center of Laboratory Animal Science at South China Agricultural University. The experiment involved healthy adult British shorthair cats, and all experimental animals were individually raised in a dedicated room in the South China Agricultural University Laboratory Animal Center. All animals were kept and housed in compliance with the applicable regulations of the Animal Welfare Act. The room where the cat lives is under a constant temperature and humidity (26 °C, 50%) with a 12 h light/dark cycle. And one cage (1.1 × 0.7 × 0.62 m) per cat was provided for rest and shelter at night. During the day, all cats were able to maintain daily interaction with people. Necessary immunizations and deworming treatments were performed prior to the experiment, and they had not been administered any medications (including antibiotics) in the month leading up to the study. Daily cleaning and disinfecting were carried out to ensure the cat enclosure remained clean.

### 2.2. Diets and Feeding

According to the nutritional recommendations of AAFCO (2017), the cats in the study were given 100 g of pet food daily to fulfill their energy requirements. The foundational diet used in this study consisted of commercially available cat food manufactured by Ramical Pet Health Technology Co., Ltd. Foshan, China The nutritional composition of the diet is presented in Table 1. GOx and MCE were provided by Guangdong VTR Bio-tech Co., Ltd. Zhuhai, China. All additives were in the form of powder and added post-spray. Throughout the experiment, the cats were granted unrestricted access to fresh water.

### 2.3. Experimental Design

A total of 24 adult British shorthair cats (mean age: 4 ± 0.21 years old; mean body weight: 4.66 ± 1.39 kg) were randomly allocated to four dietary groups. The groups included (1) a control group with a basal diet (CON group; *n* = 6, 3 male and 3 female); (2) a basal diet supplemented with 0.03% GOx (GOD group; *n* = 6, 3 male and 3 female); (3) a basal diet supplemented with 0.03% MCE (MCE group; *n* = 6, 2 male and 4 female); and (4) a basal diet supplemented with both 0.03% GOx and 0.03% MCE (GM group; *n* = 6, 4 male and 2 female). The experiment lasted a total of 56 days, including a 7 day adaptation period. Fecal scores (using the Waltham^®^ Faeces Scoring System) and quantity of food intake were recorded daily. A total fecal collection was conducted on the last four days of the trial, with fresh fecal and serum samples collected after that.

### 2.4. Blood Sample Collection and Analysis

Following an overnight fast, a 5 mL blood sample was obtained through venipuncture in the forelimb on day 49. The blood was allowed to stand for 30 min before being centrifuged at 3000 rpm at 4 °C for 15 min. After centrifugation, the supernatant was transferred to microcentrifuge tubes and stored at −80 °C for later analysis. Using the manufacturer’s instructions from commercial kits (Nanjing Jiancheng Bioengineering Institute, Nanjing, China), the serum levels of T-AOC were assessed. Additionally, serum levels of IL-2, IL-6, IL-10, IgA, and IgG were measured with commercial ELISA kits (MEIMIAN, Yancheng, China).

### 2.5. Fecal Sample Collection and Analysis

A total fecal collection was conducted over four days prior to the end of the experiment. Each morning, the feces were gathered, weighed, mixed with 10 mL of 10% hydrochloric acid per 100 g for nitrogen fixation, and then stored at −20 °C. The collected fecal samples were thoroughly mixed, air-dried, and ground for the assessment of nutrient apparent digestibility. The determination of dry matter, organic matter, crude protein, and crude fat of the feed rations and fecal samples was carried out according to GB/T6435-2014, GB/T6438-2007, GB/T6432-2018, and GB/6433-2006, respectively. Fresh fecal samples from each cat were collected within 15 min of defecation at the end of the total fecal collection and stored at −80 °C until analysis. Fecal sIgA level was measured using commercial ELISA kits (MEIMIAN, Jiangsu, China). The remaining samples were utilized for the analysis of SCFAs, BCFAs, microbiota, and metabolomics. The apparent digestibility of nutrients was calculated using a specific formula:
$$Apparent\;nutrient\;digestibility\;(\%)\;=\;\frac{\left(Nutrient\;intake-Nutrient\;in\;feces\right)}{Nutrient\;intake}\times 100\%$$

### 2.6. Analysis of Short-Chain and Branched-Chain Fatty Acids

The fecal sample was thawed and approximately 0.2 g of sample was placed into a 2 mL centrifuge tube. Then, 1 mL of ultrapure water was added, shaken, and mixed for 5 min before undergoing an ultrasonic in an ice bath for 10 min. The mixture was then centrifuged at 14,000 rpm and 4 °C for 10 min, and the supernatant was quickly transferred to a new 2 mL tube. Next, 0.25 g anhydrous sodium sulphate and 20 μL of 25% metaphosphoric acid were added and shaken to acidify and salt out the components. Then, 1 mL of methyl tert-butyl ether was added, shaken, and mixed for 5 min. Following centrifugation at 13,000 revolutions per minute (rpm) and a temperature of 4 °C for a duration of 5 min, the resultant solution was subsequently transferred to a vial and preserved at −80 °C for further analysis. Finally, the SCFAs and BCFAs were quantified using a gas chromatography–mass spectrometry (GCMS-QP2020 system; Shimadzu, Tokyo, Japan).

### 2.7. Fecal and Serum Untargeted Metabolomics Analysis

#### 2.7.1. Serum Untargeted Metabolomics Analysis

The serum sample was thawed and vortexed for 2 min. 200 μL of the serum was transferred to a sterile, enzyme-free EP tube, followed by the addition of 800 μL of chromatography-grade methanol, and vortexed for 2 min. The mixture was then centrifuged for 15 min at 14,500 rpm and 4 °C. The supernatant was dried using a vacuum centrifuge (Shenzhen, China). The residue was then reconstituted in 200 μL of chromatography-grade methanol and vortexed for 2 min. After sonication in an ice bath for 10–15 min, the sample was centrifuged at 14,500 revolutions per minute (rpm) and a temperature of 4 °C for 15 min. The solution was transferred to a vial and preserved at −80 °C until the time of analysis. An untargeted metabolomic analysis of serum was conducted using the UPLC-Orbitrap-MS/MS system manufactured by Thermo Fisher Scientific (Q-Exactive Focus, Waltham, Massachusetts, USA). Specific instrument parameters and sample processing procedures were referenced from previously published papers by the research team [27]. Further analyses were performed using MetaboAnalyst 5.0.2. https://www.metaboanalyst.ca/ (accessed on 10 September 2024) and the results were displayed graphically.

#### 2.7.2. Fecal Untargeted Metabolomics Analysis

The fecal sample was thawed and vortexed for 2 min. Approximately 60 mg of the sample was measured into a 2 mL centrifuge tube. 600 μL of methanol was added: water (1:1, *v*/*v*) and magnetic beads, and then the mixture was homogenized. Ultrasonic crushing was performed at a low temperature for 10 min and placed at −20 °C for 30 min. The samples were then centrifuged at 14,500 revolutions per minute (rpm) and a temperature of 4 °C for 15 min, and 200 μL of the supernatant was subjected to evaporation utilizing a vacuum centrifuge. Subsequently, the sample was redissolved with 200 μL of chromatographic-grade methanol and vortexed for a duration of 2 min. Following this, the sample underwent sonication in an ice bath for a period of 10 to 15 min, after which it was centrifuged at 14,500 rpm and 4 °C for 15 min. The resulting solution was transferred to a vial and stored at −80 °C in preparation for measurement. The detection protocol employed was analogous to that used for serum samples.

### 2.8. 16S rRNA Sequencing

Total bacteria DNA was extracted from fecal samples utilizing the EZNA^®^ Stool DNA Kit (D4015, Omega, Norcross, GA, USA) according to the standard procedure. Quantitative insights into microbial ecology were employed to examine the bioinformatics of sequencing data. The 16S V3–V4 rRNA region was amplified using specific primers (805R: GACTACHVGGGTATCTAATCC and 341F: CCTAYGGGRBGCASCAG), with F and R indicating forward and reverse, respectively. PCR amplification products were analyzed through 2% agarose gel electrophoresis and subsequently purified using AMPureXT beads (Beckman Coulter Genomics, Danvers, MA, USA). The purified PCR products were employed with the Quant-iT PicoGreen dsDNA Assay Kit on a Qubit fluorescence quantitative system. Paired-end sequencing was conducted on the Illumina NovaSeq 6000 platform in accordance with the manufacturer’s standard protocol. Subsequently, reads were subjected to quality filtering using QIIME quality filters (QIIME 1.9.1). The QIIME2 process was used to analyze alpha diversity (α-diversity) and beta diversity (β-diversity). Bioinformatic analysis was performed using the OmicStudio tools at https://www.omicstudio.cn/tool (accessed on 26 September 2024). Linear discriminant analysis (LDA) and effect size (LEfSe) were processed by using the LDA score of >1.5 using LEfSe software (2025).

### 2.9. Statistical Analysis

Statistical analysis was performed using SPSS version 26.0, while GraphPad Prism version 8.0.3 was employed for the generation of graphical representations. The Shapiro–Wilk test was used to assess normality, and the F-test was used to examine homogeneity of variance. Differences between multiple groups were analyzed by one-way ANOVA for between-group differences, and LSD was used for post hoc tests. To compare fecal microbial bacterial diversity, Student’s *t*-test was applied; FDR-corrected *p* < 0.05 and 0.05 ≤ *p* < 0.10, indicating statistically significant differences and trends, respectively. Data are presented as mean ± standard error (SE). Additionally, variable importance in the projection (VIP) was assessed using the OPLS-DA model, with metabolites exhibiting VIP values greater than 1 and *p*-values less than 0.05 classified as differential metabolites. The KEGG database was used for the functional annotation of these differential metabolites, which were subsequently mapped to the KEGG pathway database via MetaboAnalyst version 5.0.2. All collected samples for microbiota and metabolomics analysis were included in the final dataset. No samples were excluded as outliers.

## 3. Results

### 3.1. Effects of GOx and MCE on Body Weight and Apparent Digestibility

According to Table 2, the apparent digestibility of crude protein in GM group decreased (*p* < 0.05), while the apparent digestibility of crude fat, organic matter, dry matter, and crude fiber remained largely unchanged (*p* > 0.05). Furthermore, there were no effects on the apparent digestibility of any nutrient in the GOD and MCE groups (*p* > 0.05). The GM group experienced weight loss. However, GOx and MCE had not significantly affected body weight and food intake in the four groups at the conclusion of the experiment.

### 3.2. Effects of GOx and MCE on SCFAs and BCFAs

Table 3 shows that dietary supplementation with GOx and MCE had no effects on SCFAs or BCFAs concentrations (*p* > 0.05).

### 3.3. Effects of GOx and MCE on Immunity, Antioxidant Capacity, and Inflammatory Factors

Compared to CON group, the GOD group showed an increase in T-AOC and sIgA levels while significantly reducing IgA and IL-6 levels. The MCE group increased the levels of IgA, IgG, and fecal sIgA, and the levels of IL-2 were reduced. Meanwhile, the GM group exhibited an increase in IgG and IL-2 levels, accompanied by a decrease in IL-10 levels (Figure 1).

### 3.4. Effects of GOx and MCE on Serum Metabolome

The volcano plots revealed the distribution of 633 metabolites identified in the serum of the CON, GOD, MCE, and GM groups. Compared to the CON group, the GOD group up-regulated 9 metabolites and down-regulated 18 metabolites (Figure 2a). Similarly, there were 3 up-regulated metabolites in the MCE group along with 18 down-regulated metabolites (Figure 2b). Additionally, 21 metabolites were identified as potential markers in the GM groups, with 3 up-regulated and 18 down-regulated (Figure 2c). In addition to differential metabolites, we further explored the effects of GOx and MCE on major metabolic pathways. The GOD group mainly affected taurine and hypotaurine metabolism, sulfur metabolism, and cysteine and methionine metabolism pathways (Figure 2d). The MCE group mainly affected linolenic acid metabolism, biosynthesis of histidine and purine alkaloids, and biosynthesis of unsaturated fatty acids pathways (Figure 2e). And the GM group mainly affected α-linolenic acid metabolism, peptidoglycan biosynthesis, purine metabolism, biosynthesis of histidine and purine alkaloids and linoleic acid metabolism pathways (Figure 2f).

### 3.5. Effects of GOx and MCE on Fecal Metabolome

The volcano plots illustrated the distribution of 1223 metabolites identified in the feces of the CON, GOD, MCE, and GM groups. Among them, the GOD group up-regulated 8 metabolites and down-regulated 16 metabolites compared to the CON group (Figure 3a). There were 45 differential metabolites identified in MCE groups, with 10 metabolites up-regulated and 35 metabolites down-regulated (Figure 3b). Additionally, 24 metabolites were identified as potential biomarkers in GM groups, with 12 metabolites up-regulated and 12 metabolites down-regulated in the GM group (Figure 3c). Apart from the differential metabolites, we further explored the effects of GOx and MCE on major metabolic pathways. As shown in Figure 3d, the GOD group mainly affected phenylpropanoid biosynthesis and phenylalanine metabolism pathways. The MCE group mainly affected phenylalanine, tyrosine, and tryptophan biosynthesis, tryptophan metabolism, and phenylalanine metabolism pathways (Figure 3e). The GM group mainly affected tryptophan metabolism, phenylpropanoid biosynthesis, and phenylalanine metabolism pathways (Figure 3f).

### 3.6. Effects of GOx and MCE on Fecal Microbiome

Alpha diversity analysis of microorganisms in the feces of four groups was carried out in this experiment. Compared with the CON group, the Simpson index was lower in the GOD group, while there was no difference in abundance Choa1 number, Shannon index, and obseved-otus number (Figure 4). In addition, the GM and MCE groups did not affect Simpson index, abundance Choa1 number, Shannon index, and obseved-otus index. These results suggest that GOx and MCE have little effect on the Alpha diversity of the cat gut microbiome. The principal component analysis (PCA) results of the four groups are shown in Figure 4e. None of the GM, GOD, and MCE groups were distinctly separated from the CON group. The principal coordinate analysis (PCoA) results of the four groups are shown in Figure 4f. The GM group exhibited some separation from the CON group, while the GOD and MCE groups showed remarkable separation from the CON group. These results indicated that there were significant differences in the gut microbiome among the four groups.

The distribution of the gut microbiome at the phylum level is shown in Figure 5a. It can be observed from the figure that the dominant phylum in the intestinal tract of the four groups was Firmicutes, Actinobacteriota, Proteobacteria, and Bacteroidota. Differential bacteria at phylum and genus levels are shown in Figure 5b. Compared to the CON group, the number of Bacteroidota (phylum) decreased and *Subdoligranulum* (genus) increased in the GM group (*p* < 0.05). Firmicutes (phylum) and *Cetobacterium* (genus) increased, and Actinobacteriota (phylum), *Desulfovibrio* (genus), and *Klebsiella* (genus) decreased in the GOD group (*p* < 0.05), and the number of *Cetobacterium* (genus), *Desulfovibrio* (genus), *Escherichia-Shigella* (genus), and *Klebsiella* (genus) decreased in the MCE group.

In this experiment, LEfSe analysis (LDA > 3.5) was used to further identify differential bacteria. A total of 18 taxonomic biomarkers were identified when comparing the CON and GOD groups. The CON group demonstrated a significant increase in the relative abundance of several taxa, including Actinobacteriota (phylum), Coriobacteriaceae (family), *Phascolarctobacterium* (genus), *Mycoplasma* (genus), *Blautia* (genus), and *Collinsella* (genus), while the GOD group exhibited a significant increase in the relative abundance of Firmicutes (phylum) and *Hoyosella* (genus) (Figure 5c). Twelve taxonomic biomarkers were present between the CON and MCE groups, with the CON group increasing the relative abundance of Enterobacteriaceae (family), Lactobacillaceae (family), and *Escherichia_Shigella* (genus). Conversely, the MCE group increased the relative abundance of Morganellaceae (family) and *Arsenophonus* (genus) (Figure 5c). And 20 taxonomic biomarkers were present between the CON and GM groups, with an increase in the relative abundance of Bacteroidota (phylum), Prevotellaceae (family), *Blautia* (genus), and *Prevotella_9* (genus) in the CON group, and the GM group significantly increased the relative abundance of Ruminococcaceae (family), Reyranellaceae (family), *Subdoligranulum* (genus), *Mycobacterium* (genus), and *Reyranella* (genus) (Figure 5c).

### 3.7. Correlation of Gut Microbes with Inflammatory Factor and Immune Markers

To further explore the relationship between gut microbes and inflammatory factors and immune markers in cats, Spearman’s correlation analysis was performed (Figure 6). The results showed that *p*_Firmicutes was positively correlated with IgA. *p*_Bacteroidota was positively correlated with IgA. *g_Desulfovibrio* was negatively correlated with T-AOC, IgG, and sIgA, and positively correlated with IL-2 and IL-6. *g_Escherichia-Shigella* was negatively correlated with T-AOC, IgG, and IL-10, and positively correlated with IL-2. *g_Klebsiella* was negatively correlated with T-AOC, IgG, and sIgA, and positively correlated with IL-2.

## 4. Discussion

Numerous studies have demonstrated that GOx and MCE play crucial roles in economic animals, enhancing the production performance and health of livestock [28,29,30]. This study aims to provide data supporting the use of GOx and MCE in pet food by investigating their effects on antioxidant levels, immunity, and microbiota composition in cats.

Intact intestinal barrier function is crucial for maintaining animal health [31]. Cytokines released by immune cells serve as indicators of immune health and intestinal barrier integrity [32]. Among these cytokines, both pro-inflammatory and anti-inflammatory factors are frequently utilized as markers of intestinal inflammation [33]. Interleukin-2 (IL-2) is integral to immune modulation and cellular proliferation [34], as it promotes the secretion of IFN-γ and TNF-α, which then trigger inflammatory responses. IL-6 exhibits both pro-inflammatory and anti-inflammatory properties [35], rapidly expressed during inflammatory processes, and serves as a potential biomarker for inflammation [35,36]. IL-10 is an immunomodulatory cytokine generated by multiple cell types with potent anti-inflammatory and immune-suppressive effects, and its deficiency may cause autoimmune disease [37,38]. Immunoglobulin proteins, essential for humoral immunity, primarily consist of IgA, IgM, and IgG [39]. sIgA consists of two linked IgA monomers and is the major mucosal antibody in mammals [40], which can neutralize toxins, limit the invasion of foreign pathogenic microorganisms, and regulate gut microbiome [41,42]. Recent research has indicated that MCE can reduce IL-6 levels in pigs [43] and enhance IgG and sIgA levels in neonatal pigs [14]. We observed significantly elevated levels of IgA, IgG, and sIgA, alongside reduced levels of IL-6 and IL-2 in the MCE group, suggesting that the incorporation of MCE may bolster immunity and diminish inflammation in felines. Oxidative stress is known to induce oxidative DNA damage and abnormal protein expression, contributing to various diseases [44,45]. T-AOC levels are utilized to evaluate the body’s antioxidant capacity [46]. Studies have indicated that GOx enhances immune function and alleviates oxidative stress [46,47]. A decrease in MDA and an increase in SOD were observed in white-feathered broilers [48], aligning with the findings of the current experiment. Additionally, we observed up-regulation of eicosapentaenoic acid (EPA) in the GOD and MCE groups. EPA is an omega-3 polyunsaturated fat that inhibits inflammatory responses and relieves oxidative stress, consistent with the above results. However, the concomitant addition of GOx and MCE resulted in increased IL-2 levels and decreased IL-10 levels, suggesting that joint use of GOx and MCE may indicate a pro-inflammatory trend.

Gut microbiome is like a functional organ [49], and its composition and status significantly affect host health and disease [50]. Various factors, including diet, age, disease, and structural modifications within the gut, can affect the microbiota’s composition [49,50,51]. Our investigation demonstrated that the fecal microbiome of cats was predominantly composed of *Firmicutes*, *Actinobacteria*, *Bacteroidota,* and *Proteobacteria*, corroborating findings from prior research [49,52,53]. PCoA results based on unweighted UniFrac distances revealed distinct separations among the GOD, MCE, and CON groups, suggesting that GOx and MCE can modulate the gut microbiome of cats. Notably, increases in *Desulfovibrio*, *Escherichia-Shigella,* and *Klebsiella* were found to be detrimental to the homeostasis of the gut microbiome [54,55,56]. Hephaestin-like 1 (HEPHL1), a newly identified multicopper oxidase, plays a key role in the maintenance of iron homeostasis [57]. Research has indicated a strong negative correlation between *Desulfovibrio* and HEPHL1 [58]. Our results revealed that GOx supplementation increased the relative abundance of *Cetobacterium*, decreased the relative abundance of *Desulfovibrio* and *Klebsiella*, while MCE reduced the relative abundance of *Escherichia-Shigella*, *Desulfovibrio,* and *Klebsiella,* indicating improved gut health. *Bacteroidota*, recognized for their role in promoting intestinal health and immunity [46], and the ratio of *Firmicutes* to *Bacteroidetes,* are commonly used to measure the host’s health [59]. However, the abundance of Bacteroidota was markedly reduced in the GM group, demonstrating that the concurrent administration of GOx and MCE may adversely affect feline health. Furthermore, the LEfSe results revealed that the relative abundances of *Ruminococcaceae*, *Reyranellaceae*, *Subdoligranulum*, and *Mycobacterium* in the GM group. *Mycobacterium* is capable of infecting mammalian hosts [60] and is characterized by both latent and active infections. It has also been suggested that feline lymphadenitis may be associated with *Mycobacterium* infection [61]. Previous studies have established a negative correlation between *Bacteroidota* and disease occurrence, which may contribute to inflammatory processes [62,63]. Alterations in the abundance of specific microbiota can impact host health; changes in *Bacteroidota* and *Mycobacterium* may drive the inflammatory response observed in the GM group.

Metabolomics refers to the comprehensive analysis of endogenous metabolites that define the metabolic pathways present within biological systems [64]. These substances are typically secreted within the gastrointestinal tract and subsequently transported across the intestinal barrier into the circulatory system, where they serve as crucial regulators of host metabolism [3]. These metabolites offer valuable information regarding the health condition of the gastrointestinal tract and the general well-being of the organism [65]. Notably, gut microbial metabolites are intricately linked to nutritional status, health, and metabolic processes [66]. In the present study, metabolomic analyses were conducted to investigate the potential mechanisms through which GOx and MCE influence feline metabolism. Glutathione (GSH) plays a critical role in alleviating oxidative damage and maintaining redox homeostasis [67,68]. γ-glutamylcysteine (γ-GC) is a precursor of GSH with antioxidant and anti-inflammatory properties [69]. A previous study has shown that γ-GC can protect against alcoholic liver disease through suppressing the effects of oxidation and alleviating inflammation [70]. In addition, γ-GC can attenuate inflammatory responses in BALB/c mice [68]. In the present study, up-regulation of γ-GC levels explained the anti-inflammatory and oxidative stress-relieving effects of GOx and MCE, which is also consistent with the serum-related results. Remarkably, the up-regulation of quinolinic acid in feces has been linked to neurological disorders [71]. Therefore, we speculated that the simultaneous addition of GOx and MCE could increase inflammation and the potential for disease in cats. Linoleic acid is a vital fatty acid for felines, contributing positively to the health of their skin and coat [72]. Research has indicated that linoleic acid metabolism represents the sole pathway exhibiting differential expression within the healthy groups, characterized by notably elevated enrichment scores in comparison to the diseased group [73]. Tryptophan metabolites are involved in maintaining normal barriers, immune responses, and tissue repair [74,75]. Our study demonstrated significant enrichment of linoleic acid and tryptophan metabolism in both the GM and MCE groups, suggesting that adding MCE in the diet is advantageous for feline health. Cats cannot synthesize sufficient taurine on their own and must obtain it from their diet [76]. It is an essential nutrient for maintaining retinal health, normal heart muscle function, bile acid binding, and fetal development [77,78]. While GOD may provide a potential auxiliary regulatory role in maintaining taurine levels by influencing the taurine metabolic pathway. Moreover, several studies have revealed that changes in amino acid levels can substantially influence the composition of microbial communities (e.g., *Proteobacteria*) [79,80]. Our findings revealed changes in *Proteobacteria* within the GOx and MCE groups, reflecting the interaction between amino acid metabolism and gut microbiota. This is consistent with previous studies in piglets [81]. We identified metabolic pathways in both fecal metabolites and serum metabolites that are advantageous for British shorthair cats. Our findings indicated that fecal metabolites primarily influence gut microbes and amino acid metabolism, while serum metabolites are mainly associated with lipid-related metabolic pathways.

GOx and MCE show potential application value in alleviating oxidative stress and enhancing immunity. However, this study is limited by the small sample size of only 24 cats. Additionally, due to the unique characteristics of the feline species, we were unable to conduct further mechanism-related research (such as genetic testing). Nevertheless, future studies involving a larger number of cats to provide more data for pet food development remain worthy of attention.

## 5. Conclusions

In conclusion, our study indicated that GOx and MCE may mitigate oxidative stress and enhance immune function in cats through the regulation of cytokine levels, specifically T-AOC, sIgA, IgA, and IgG. Additionally, these additives may influence gut microbiota and metabolites, consequently promoting intestinal health in felines. However, the combination of the two additives may have some negative effects on the inflammatory response. Further exploration is required to gain a deeper understanding of the potential mechanisms through which these substances influence feline intestinal health, as well as to determine the optimal dosages for their application.

## Figures and Tables

**Figure 1 metabolites-15-00759-f001:**
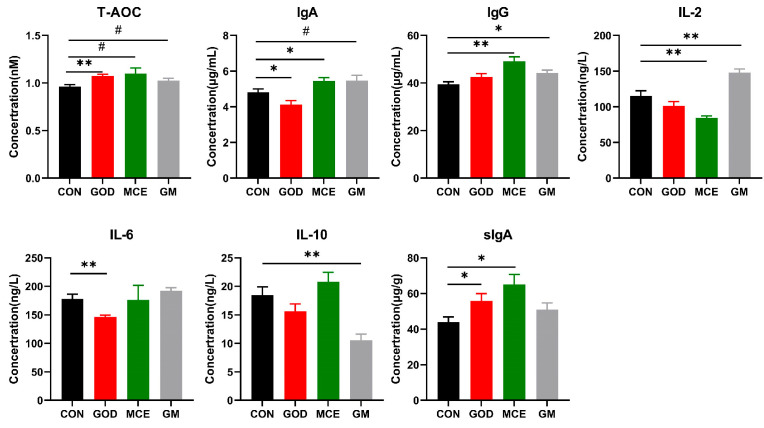
Effects of GOx and MCE on immunity, antioxidant capacity, and inflammatory factors. Data are presented as mean ± SE. The symbol (*) denotes statistically significant differences (* *p* < 0.05), the symbol (**) denotes statistically highly significant differences (** *p* < 0.01), and the symbol (#) represents difference tendency (# 0.05 ≤ *p* ≤ 0.10). T-AOC: total antioxidant capacity; IgA: immunoglobulin A; IgG: immunoglobulin G; IL-2: interleukin 2; IL-6: interleukin 6; IL-10: interleukin 10; sIgA: fecal secretory immunoglobulin A; CON: basal diet group; GOD: basal diet + 0.03% GOx group; MCE: basal diet + 0.03% MCE group; GM: basal diet + 0.03% GOx and 0.03% MCE group.

**Figure 2 metabolites-15-00759-f002:**
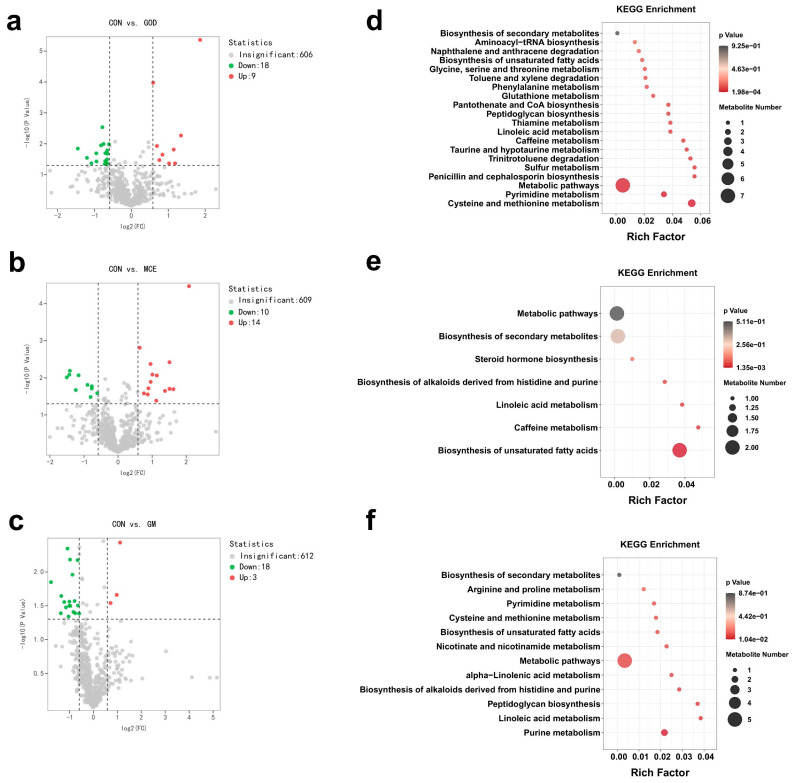
Effects of GOx and MCE on serum metabolome of cats. (**a**) Volcano plots were used to compare the differential metabolite between CON and GOD groups. (**b**) Volcano plots were used to compare the differential metabolite between CON and MCE groups. (**c**) Volcano plots were used to compare the differential metabolite between CON and GM groups. (**d**) Enrichment analysis of metabolic pathways in the CON and GOD groups. (**e**) Enrichment analysis of metabolic pathways in the CON and MCE groups. (**f**) Enrichment analysis of metabolic pathways in the CON and GM groups. CON: basal diet group; GOD: basal diet + 0.03% GOx group; MCE: basal diet + 0.03% MCE group; GM: basal diet + 0.03% GOx and 0.03% MCE group.

**Figure 3 metabolites-15-00759-f003:**
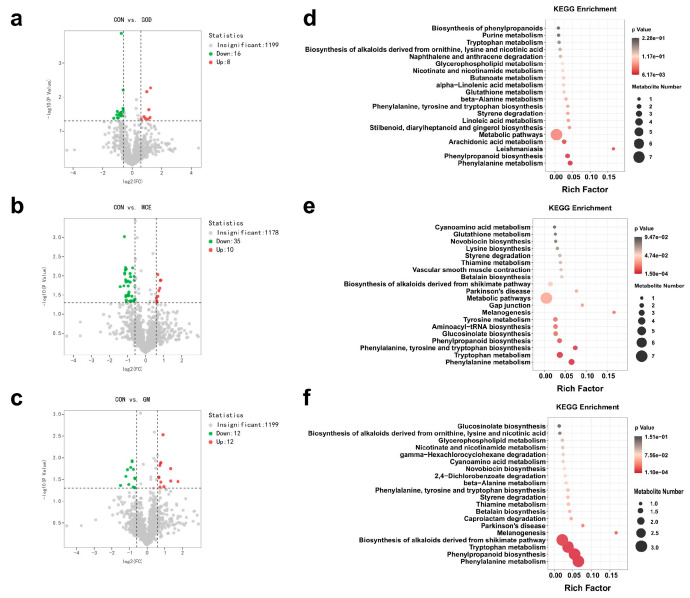
Effects of GOx and MCE on fecal metabolome of cats. (**a**) Volcano plots were used to compare the differential metabolite between CON and GOD groups. (**b**) Volcano plots were used to compare the differential metabolite between CON and MCE groups. (**c**) Volcano plots were used to compare the differential metabolite between CON and GM groups. (**d**) Enrichment analysis of metabolic pathways in the CON and GOD groups. (**e**) Enrichment analysis of metabolic pathways in the CON and MCE groups. (**f**) Enrichment analysis of metabolic pathways in the CON and GM groups. CON: basal diet group; GOD: basal diet + 0.03% GOx group; MCE: basal diet + 0.03% MCE group; GM: basal diet + 0.03% GOx and 0.03% MCE group.

**Figure 4 metabolites-15-00759-f004:**
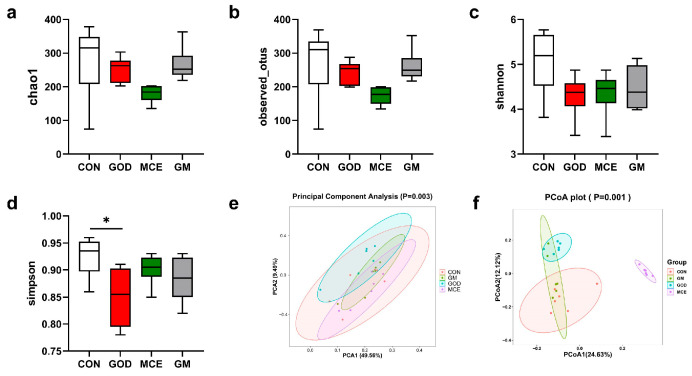
(**a**) Chao1 number; (**b**) obseved-otus number; (**c**) Shannon index; (**d**) Simpson index; (**e**) PCA analysis of bacteria in the feces of cats; (**f**) PCoA analysis of bacteria in the feces of cats in four groups. CON: basal diet group; GOD: basal diet + 0.03% GOx group; MCE: basal diet + 0.03% MCE group; GM: basal diet + 0.03% GOx and 0.03% MCE group.The symbol (*) denotes statistically significant differences (* *p* < 0.05).

**Figure 5 metabolites-15-00759-f005:**
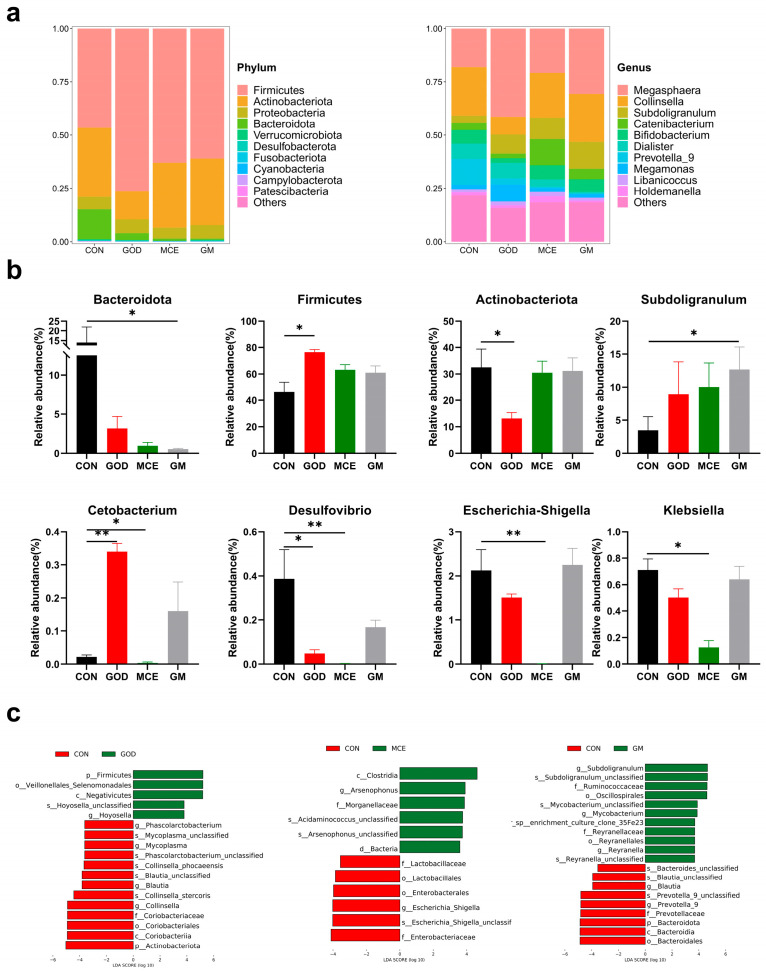
(**a**) Column stacked plots analysis of microbial composition at phylum and genus levels. (**b**) Differential bacteria in the feces of four groups. (**c**) LEfSe analysis of fecal microorganisms between CON and GOD groups. CON: basal diet group; GOD: basal diet + 0.03% GOx group; MCE: basal diet + 0.03% MCE group; GM: basal diet + 0.03% GOx and 0.03% MCE group. The symbol (*) denotes statistically significant differences (* *p* < 0.05), the symbol (**) denotes statistically highly significant differences (** *p* < 0.01).

**Figure 6 metabolites-15-00759-f006:**
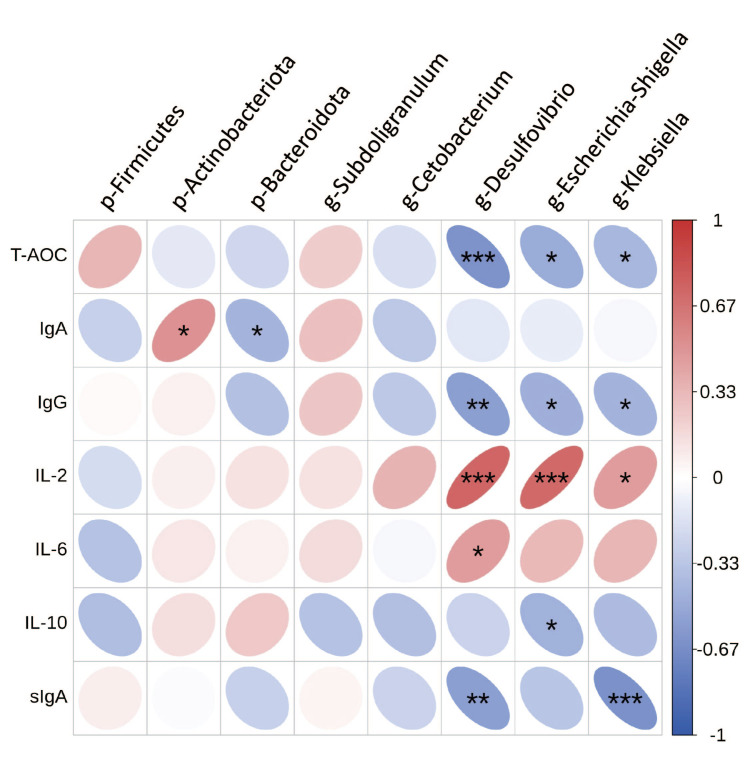
Correlation of gut microbes with inflammatory factor and immune markers. The symbol (*) denotes statistically significant differences (* *p* < 0.05), the symbol (**) denotes statistically highly significant differences (** *p* < 0.01), and the symbol (***) represents extremely significant differences (*** *p* < 0.001).

**Table 1 metabolites-15-00759-t001:** Ingredients and nutrient levels of the experimental diets.

Items			
Ingredients (%)		Nutrient levels ^2^ (%)	
Corn	27.00	DM	92.61
Wheat	10.00	CP	31.21
Flours	8.00	CF	10.93
Corn gluten meal	8.00	OM	91.35
Soybean meal	8.00		
Beef bone meal	5.00		
Chicken meat	15.00		
Sugar beet pulp	3.00		
Calcium hydrogen Phosphate	1.00		
Zeolite powder	1.00		
Premix ^1^	2.00		
Mixed oils	7.00		
Composite seasoning	5.00		
Total	100		

^1^ Vitamin and mineral premix provided the following kg^−1^ diet: vitamin D3, vitamin A, vitamin B1 (thiamine), vitamin B6, vitamin B2 (riboflavin), vitamin E, vitamin K3, vitamin B12, niacin, biotin, folacin, calcium pantothenate, choline chloride, Zn, Fe, Mn, Cu, I, Se. ^2^ Measured values in dry matter basis.

**Table 2 metabolites-15-00759-t002:** Effects of GOx and MCE on body weight and apparent digestibility.

	CON	GOD	MCE	GM
Body weight change (kg)	−0.07 ± 0.11	0.01 ± 0.21	0.03 ± 0.14	−0.17 ± 0.17
Food intake (g)	54.10 ± 5.28	63.54 ± 5.26	57.31 ± 4.67	58.22 ± 6.22
Fecal score	2.47	2.49	2.50	2.50
Digestibility				
Dry matter (%)	81.74 ± 1.29	79.77 ± 1.30	81.30 ± 2.25	78.94 ± 0.99
Crude protein (%)	85.49 ± 0.90	84.14 ± 0.75	85.43 ± 1.76	82.22 ± 0.99 *
Crude fat (%)	94.91 ± 0.97	92.74 ± 1.48	96.37 ± 0.71	93.46 ± 1.11
Organic material (%)	85.68 ± 0.91	84.23 ± 0.92	85.27 ± 1.66	83.50 ± 0.78
Crude fiber (%)	37.99 ± 6.85	29.45 ± 11.20	44.03 ± 7.96	18.07 ± 10.73

CON: basal diet group; GOD: basal diet + 0.03% GOx group; MCE: basal diet + 0.03% MCE group; GM: basal diet + 0.03% GOx and 0.03% MCE group. Data are presented as mean ± SE. * Indicates a statistically significant difference (*p* < 0.05) was found when digestibility was compared with the CON group.

**Table 3 metabolites-15-00759-t003:** Effects of GOx and MCE on SCFAs and BCFAs.

Items (ng/g)	CON	GOD	MCE	GM
Acetic acid	1282.02 ± 93.09	1350.31 ± 49.16	1434.59 ± 43.79	1439.03 ± 43.12
Propanoic acid	781.02 ± 84.43	769.12 ± 128.15	856.10 ± 120.20	799.17 ± 115.22
Isobutyric acid	26.71 ± 4.47	29.88 ± 9.18	20.77 ± 5.16	29.55 ± 8.38
Butyric acid	428.16 ± 57.42	383.46 ± 73.24	402.8 ± 42.55	464.21 ± 25.51
Isovaleric acid	54.44 ± 7.64	62.88 ± 17.00	44.16 ± 10.24	60.54 ± 16.41
Valeric acid	242.95 ± 70.53	266.65 ± 57.56	252.8 ± 48.67	292.00 ± 47.23

CON: basal diet group; GOD: basal diet + 0.03% GOx group; MCE: basal diet + 0.03% MCE group; GM: basal diet + 0.03% GOx and 0.03% MCE group. Data are presented as mean ± SE.

## Data Availability

The data presented in this study are available in the manuscript.

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
