# Peer review of "Effects of Glucose Oxidase and Macleaya cordata Extract on Immune Function, Antioxidant Capacity, and Gut Microbiota in British Shorthair Cats"

_metabolites, 2025, doi:10.3390/metabo15120759_

Round 1
Reviewer 1 Report
Comments and Suggestions for Authors
This manuscript investigates the effects of dietary supplementation with glucose oxidase (GOx) and Macleaya cordata extract (MCE) on immune responses, antioxidant capacity, gut microbiota composition, and metabolic profiles in domestic cats. The study addresses a relevant and relatively unexplored topic, as most of the literature on these additives focuses on livestock species. The work is methodologically sound, the data are presented clearly, and the multi-omics approach (immunology, microbiota, serum and fecal metabolomics) adds depth to the findings. Importantly, the study highlights potential antagonistic effects when GOx and MCE are combined, which is scientifically interesting and practically relevant for the pet nutrition industry. Overall, the study is of good quality and offers novel insights, but there are several areas where the clarity, rigor, and interpretation could be strengthened to support the conclusions more robustly.
Major Comments
Integration of immunological and microbial findings (Fig. 1 and 3).
The observed increase in T-AOC and sIgA, together with the reduction of IL-6 in the GOx group, aligns with shifts in microbial composition (notably decreased Desulfovibrio and Klebsiella). This relationship is central to the authors’ argument but is currently only implicit. A clearer discussion connecting these immunological outcomes to specific microbial shifts, ideally supported by statistical interaction or mediation analysis, would make the interpretation more compelling.
Antagonistic interaction in the combined treatment (GM). The data consistently show that the combination of GOx and MCE fails to enhance beneficial effects and may even reverse some of them (e.g., decreased protein digestibility, increased IL-2, reduced IL-10, altered microbiota with an increase in Mycobacterium). This is a key result, but is not sufficiently explored mechanistically. The authors should elaborate on potential explanations (e.g., competition in redox signaling, additive instability, or microbiota interference).
Statistical rigor in microbiota analysis. While PCoA plots demonstrate group separation, no formal statistical test (e.g., PERMANOVA/Adonis) is reported. Including such tests would strengthen claims of significant community shifts. Similarly, clarification on multiple-testing correction for LEfSe or correlation analyses is necessary to avoid overinterpretation of marginal signals.
Metabolomic interpretation (Figs. 2 and 4). The metabolomic data are suggestive but underdeveloped. The discussion should specify key metabolites driving pathway enrichment, not just the pathways themselves. This is particularly relevant for the taurine/cysteine axis in serum and the phenylpropanoid/triptophan pathways in feces, which are mechanistically linked to mucosal immunity and microbiota–host cross-talk.
Causality vs. correlation (Fig. 5). The correlation network between taxa and immune/metabolic parameters is informative but should be presented with appropriate caution. The manuscript would benefit from an explicit statement acknowledging that these are associations, not evidence of direct causality.
Minor Comments
-
The legends of Fig. 1 should specify clearly which measurements were obtained from serum vs. fecal samples (e.g., sIgA vs. IgA).
-
P-values or confidence intervals should be provided for key comparisons (particularly digestibility and immune markers).
-
The authors should include, in supplementary materials, full lists of differential metabolites with fold-changes and statistical values.
-
Methods should specify how outliers, if any, were handled in microbiota and metabolomic datasets.
-
The discussion should acknowledge the relatively small sample size (n = 6 per group) and its implications for statistical power.
Conclusion
This study provides novel and potentially impactful data on the immunometabolic and microbial effects of GOx and MCE supplementation in cats. The experimental design is appropriate and the data are of reasonable quality. However, to fully support the authors’ conclusions, the manuscript would benefit from stronger mechanistic interpretation, additional statistical detail, and clearer integration of multi-omics findings. Addressing these points will substantially improve the scientific depth and credibility of the work.
Author Response
1.Integration of immunological and microbial findings (Fig. 1 and 3). The observed increase in T-AOC and sIgA, together with the reduction of IL-6 in the GOx group, aligns with shifts in microbial composition (notably decreased Desulfovibrio and Klebsiella). This relationship is central to the authors’ argument but is currently only implicit. A clearer discussion connecting these immunological outcomes to specific microbial shifts, ideally supported by statistical interaction or mediation analysis, would make the interpretation more compelling.
Response: To further explore the relationship between gut microbes and inflammatory factor and immune markers in cats, Spearman’s correlation analysis was performed (Fig. 6; in part 3.7 ).
2.Statistical rigor in microbiota analysis. While PCoA plots demonstrate group separation, no formal statistical test (e.g., PERMANOVA/Adonis) is reported. Including such tests would strengthen claims of significant community shifts. Similarly, clarification on multiple-testing correction for LEfSe or correlation analyses is necessary to avoid overinterpretation of marginal signals.
Response: Modified. Part 2.8
3.Metabolomic interpretation (Figs. 2 and 4). The metabolomic data are suggestive but underdeveloped. The discussion should specify key metabolites driving pathway enrichment, not just the pathways themselves. This is particularly relevant for the taurine/cysteine axis in serum and the phenylpropanoid/triptophan pathways in feces, which are mechanistically linked to mucosal immunity and microbiota–host cross-talk.
Response :The number of differential metabolites involved is too large, so our discussion of the metabolomics results only covers a subset of key metabolites (such as γ-glutamylcysteine and quinolinic acid). Line 398-403.
4.Causality vs. correlation (Fig. 5). The correlation network between taxa and immune/metabolic parameters is informative but should be presented with appropriate caution. The manuscript would benefit from an explicit statement acknowledging that these are associations, not evidence of direct causality.
Response: Modified. In the Results section describing Figure 5, we have carefully revised our wording throughout to avoid any implication of causality. Terms such as "causes" or "leads to" have been replaced with more precise language like "is associated with," "correlates with," or "co-varies with." Part 3.7.
5.The legends of Fig. 1 should specify clearly which measurements were obtained from serum vs. fecal samples (e.g., sIgA vs. IgA).
Response: Modified.
6.P-values or confidence intervals should be provided for key comparisons (particularly digestibility and immune markers).
Response: Digestibility results are annotated with specific p-values, while differences in immune biomarkers are indicated by symbols such as *, with further clarification provided in the figure captions.
7.The authors should include, in supplementary materials, full lists of differential metabolites with fold-changes and statistical values.
Response: The data for that section has been supplemented.
8.Methods should specify how outliers, if any, were handled in microbiota and metabolomic datasets.
Response: Added. All collected samples for microbiota and metabolomics analysis were included in the final dataset. No samples were excluded as outliers.
9.The discussion should acknowledge the relatively small sample size (n = 6 per group) and its implications for statistical power.
Response: The sample size for this trial was determined based on resource availability (e.g., scarcity of specific cats) and in accordance with ethical principles to minimize animal use, but it was not formally validated through prior efficacy calculations. We sincerely acknowledge that we failed to conduct a formal pre-specified efficacy analysis for the primary outcome measures (such as T-AOC or sIgA).
Reviewer 2 Report
Comments and Suggestions for Authors
Dear Authors,
the manuscript explores an interesting and under-investigated topic. The study provides potentially valuable insights into the nutritional modulation of immunity and gut microbiota in felines, an area with limited scientific coverage. However, several methodological, analytical, and interpretative issues need to be addressed to strengthen the manuscript. I suggest revising the entire manuscript carefully and to add the line numbers.
Abstract
The Abstract is overly detailed but lacks focus on objectives, key findings, and implications. Please condense to 200–250 words, include main quantitative outcomes ( % changes, p-values), and avoid overstatements (“prove”, “can enhance”). Replace with more cautious language as “may enhance”, “suggest potential for”. Clarify whether GOx and MCE acted synergistically or antagonistically. Replace “Abstracts” with “Abstract”.
Introduction
The Introduction provides an extensive background on Glucose oxidase and Macleaya cordata extract in livestock, but the novelty in relation to cats remains insufficiently emphasized. The rationale should clearly explain why these additives might be beneficial in felines and highlight the specific gap in current knowledge. Consider explicitly stating that feline digestive physiology differs substantially from omnivorous species, and thus extrapolation from pigs or chickens is limited. The aim should be stated more directly at the end of the Introduction. The Introduction heavily relies on studies performed in livestock species (pigs, broilers, dairy cows). While these references are relevant for contextualizing the biological activity of GOx and MCE, they do not directly support the rationale for testing these additives in cats. The authors should add references specific to companion animals, or at least acknowledge physiological differences that limit extrapolation from livestock to felines. Simplify “have become popular as eco-friendly feed additives within the realm of animal production” to “have become popular eco-friendly feed additives in animal nutrition”.
Materials and Methods
The study design is generally appropriate, but some aspects would benefit from clarification. The use of 24 cats (6 per group) appears limited given the number of endpoints assessed (immune, antioxidant, metabolomic, and microbiota parameters). It would be helpful if the authors could justify this sample size statistically or acknowledge it as a limitation at the end of the Discussion. In addition, the sex distribution among groups seems unbalanced (GM group: 4 males and 2 females), which could potentially influence immune and metabolic outcomes. Please clarify whether randomization accounted for sex or body weight to minimize such confounding effects. For transparency and reproducibility, I recommend specifying the exact number of biological replicates analyzed per group and indicating whether any samples were pooled. Further, please provide more detail on the microbiota analysis and how metabolomic data were normalized and scaled prior to statistical analysis. Finally, the Methods should include the ethical approval number and the approving institution. Indicate the adaptation period before the 49-day feeding trial.
Methodological clarity
The methodological section is overly detailed in describing laboratory steps but lacks essential validation information. Please specify whether commercial ELISA kits were validated for feline samples, as cross-reactivity. For GC-MS and metabolomics, provide details about calibration standards, internal controls, and reproducibility. Clarify whether both positive and negative ionization modes were analyzed. The formula for nutrient digestibility should indicate units and specify whether fecal nutrients were corrected for endogenous losses.
Statistical analysis
The statistical approach would benefit from a clearer description. Please indicate whether data normality and homogeneity of variance were tested before applying ANOVA, and specify the software and database used for the 16S rRNA analysis. If multiple comparisons were performed in metabolomic or microbiota analyses, please clarify whether any correction was applied.
Results
Some results appear inconsistent, for example the GOD group shows both an increase in sIgA and a decrease in IgA levels, which is biologically improbable. Please verify the dataset and figure labels. The reduction in protein digestibility and weight loss in the GM group (Table 2) suggests a potential antagonistic interaction between GOx and MCE or possible effects on feed palatability; this should be discussed in the discussion section. Figures 2–3 lack specific metabolite names contributing to enriched pathways, listing key metabolites would improve interpretation.
Discussion
The Discussion section is largely descriptive and occasionally repetitive. A more integrated and mechanistic organization would improve readability for example, grouping findings according to immune modulation, oxidative stress responses, and microbiota-metabolome interactions. Statements implying causality “joint use of GOx and MCE may exert adverse effects” should be expressed more cautiously as “may suggest pro-inflammatory tendencies”. The paragraph discussing possible Mycobacterium infection risk appears speculative and is not supported by analysis and data so, it should be either removed or clearly presented as a hypothesis. A short paragraph acknowledging study limitations such as small sample size, lack of replication, use of a commercial diet, and limited validation of metabolomic results would strengthen the discussion.
Ethical statement and conflict of interest
The conflict of interest section reveals that two authors are employed by Guangdong VTR Bio-tech Co., Ltd., which supplied GOx and MCE. This constitutes a potential source of bias. The authors should explicitly state that the company had no role in study design, data analysis, interpretation, or decision to publish. Transparency here is essential for maintaining scientific integrity.
Figures
Check the units in table 1 and include explicit measurement units for all nutrient parameters.
References
The reference section shows several formatting inconsistencies (missing DOIs, mixed capitalization, and non-uniform journal titles). The style currently used (author–date) does not follow the numbered format required by Metabolites and should be adjusted accordingly. The discussion would benefit from the inclusion of more independent studies, especially regarding feline digestive physiology, immune function, and gut microbiota. Please also verify the accuracy of each reference (year, DOI, author list) and ensure uniform formatting.
Language and Style
The manuscript is readable but requires thorough English editing. Common issues include article misuse, repetitive expressions, and inconsistent tense. Check the entire manuscript language to ensure fluency with clear transitions and concise topic sentences, especially the Discussion. Ensure that all Latin names are italicized, and units conform to SI standards.
Kind Regards
Author Response
The revised sections of the article have been highlighted in yellow.
As references cannot be modified in the revised draft, all changes must be made in the original manuscript. We apologize for any inconvenience this may cause.
1.The Abstract is overly detailed but lacks focus on objectives, key findings, and implications. Please condense to 200–250 words, include main quantitative outcomes ( % changes, p-values), and avoid overstatements (“prove”, “can enhance”). Replace with more cautious language as “may enhance”, “suggest potential for”. Clarify whether GOx and MCE acted synergistically or antagonistically. Replace “Abstracts” with “Abstract”.
Response: Modified.
2.The Introduction provides an extensive background on Glucose oxidase and Macleaya cordata extract in livestock, but the novelty in relation to cats remains insufficiently emphasized. The rationale should clearly explain why these additives might be beneficial in felines and highlight the specific gap in current knowledge. Consider explicitly stating that feline digestive physiology differs substantially from omnivorous species, and thus extrapolation from pigs or chickens is limited. The aim should be stated more directly at the end of the Introduction. The Introduction heavily relies on studies performed in livestock species (pigs, broilers, dairy cows). While these references are relevant for contextualizing the biological activity of GOx and MCE, they do not directly support the rationale for testing these additives in cats. The authors should add references specific to companion animals, or at least acknowledge physiological differences that limit extrapolation from livestock to felines. Simplify “have become popular as eco-friendly feed additives within the realm of animal production” to “have become popular eco-friendly feed additives in animal nutrition”.
Response: We conducted a comprehensive literature search using databases such as PubMed and Web of Science with keywords including ‘glucose oxidase cats’, ‘Macleaya cordata companion animals’, and ‘bloodroot extract feline’.
The search results confirm that no published studies currently exist on the application of glucose oxidase (GOx) or Macleaya cordata extract in healthy domestic cats. We recognize this represents a critical knowledge gap in the field. Consequently, the manuscript does not include directly relevant feline literature but instead cites studies establishing biological activity in monogastric animals (such as pigs and poultry).
3.The study design is generally appropriate, but some aspects would benefit from clarification. The use of 24 cats (6 per group) appears limited given the number of endpoints assessed (immune, antioxidant, metabolomic, and microbiota parameters). It would be helpful if the authors could justify this sample size statistically or acknowledge it as a limitation at the end of the Discussion. In addition, the sex distribution among groups seems unbalanced (GM group: 4 males and 2 females), which could potentially influence immune and metabolic outcomes. Please clarify whether randomization accounted for sex or body weight to minimize such confounding effects. For transparency and reproducibility, I recommend specifying the exact number of biological replicates analyzed per group and indicating whether any samples were pooled. Further, please provide more detail on the microbiota analysis and how metabolomic data were normalized and scaled prior to statistical analysis. Finally, the Methods should include the ethical approval number and the approving institution. Indicate the adaptation period before the 49-day feeding trial.
Response: The sample size for this trial was determined based on resource availability (e.g., scarcity of specific cats) and in accordance with ethical principles to minimize animal use. Therefore, we employed randomization based on body weight and sex to maximize balance. While this method aims for balance, the inherent randomness in small sample sizes (N=6 per group) resulted in the observed minor imbalances (MCE: 2♂/4♀; GM: 4♂/2♀).
Ethical approval number and the approving institution were added.
The experiment lasted a total of 56 days, including a seven-day adaptation period.
4.The methodological section is overly detailed in describing laboratory steps but lacks essential validation information. Please specify whether commercial ELISA kits were validated for feline samples, as cross-reactivity. For GC-MS and metabolomics, provide details about calibration standards, internal controls, and reproducibility. Clarify whether both positive and negative ionization modes were analyzed. The formula for nutrient digestibility should indicate units and specify whether fecal nutrients were corrected for endogenous losses.
Response: Modified. Part 2.5 and 2.7.
5.The statistical approach would benefit from a clearer description. Please indicate whether data normality and homogeneity of variance were tested before applying ANOVA, and specify the software and database used for the 16S rRNA analysis. If multiple comparisons were performed in metabolomic or microbiota analyses, please clarify whether any correction was applied.
Response: Modified. Part 2.8 and 2.9.
6.Some results appear inconsistent, for example the GOD group shows both an increase in sIgA and a decrease in IgA levels, which is biologically improbable. Please verify the dataset and figure labels. The reduction in protein digestibility and weight loss in the GM group (Table 2) suggests a potential antagonistic interaction between GOx and MCE or possible effects on feed palatability; this should be discussed in the discussion section. Figures 2–3 lack specific metabolite names contributing to enriched pathways, listing key metabolites would improve interpretation.
Response: Modified. In discussion Section, Second Paragraph
All differential metabolites are listed in the appendix.
7.The Discussion section is largely descriptive and occasionally repetitive. A more integrated and mechanistic organization would improve readability for example, grouping findings according to immune modulation, oxidative stress responses, and microbiota-metabolome interactions. Statements implying causality “joint use of GOx and MCE may exert adverse effects” should be expressed more cautiously as “may suggest pro-inflammatory tendencies”. The paragraph discussing possible Mycobacterium infection risk appears speculative and is not supported by analysis and data so, it should be either removed or clearly presented as a hypothesis. A short paragraph acknowledging study limitations such as small sample size, lack of replication, use of a commercial diet, and limited validation of metabolomic results would strengthen the discussion.
Response: Modified.
8.The conflict of interest section reveals that two authors are employed by Guangdong VTR Bio-tech Co., Ltd., which supplied GOx and MCE. This constitutes a potential source of bias. The authors should explicitly state that the company had no role in study design, data analysis, interpretation, or decision to publish. Transparency here is essential for maintaining scientific integrity.
Response: Modified.
9.Check the units in table 1 and include explicit measurement units for all nutrient parameters.
Response: Modified.
10.The reference section shows several formatting inconsistencies (missing DOIs, mixed capitalization, and non-uniform journal titles). The style currently used (author–date) does not follow the numbered format required by Metabolites and should be adjusted accordingly. The discussion would benefit from the inclusion of more independent studies, especially regarding feline digestive physiology, immune function, and gut microbiota. Please also verify the accuracy of each reference (year, DOI, author list) and ensure uniform formatting.
Response: Modified.
Round 2
Reviewer 1 Report
Comments and Suggestions for Authors
The revised version of the manuscript represents a meaningful improvement over the original submission. I appreciate the authors’ efforts to incorporate additional analyses and clarify several interpretive points raised in the initial review. In particular, the addition of PERMANOVA testing, more precise language around associations, clearer figure legends, and expanded metabolite reporting have strengthened the manuscript substantially.
That said, there remain a few key areas that would benefit from further refinement to bring the work fully in line with the standards expected for publication. First, while the inclusion of Spearman correlation analyses provides some insight into the relationship between immune markers and microbial shifts, this approach still falls short of demonstrating any structured interaction between these domains. The manuscript would be significantly stronger with a more explicit statistical or conceptual integration of these datasets, ideally in the form of interaction testing, mediation modeling, or, at the very least, a more rigorous interpretation that goes beyond parallel description.
Second, the metabolomics section remains underdeveloped. The authors mention focusing on a few key metabolites, which is a reasonable strategy given the breadth of the data. However, the current discussion still does not clearly articulate how these metabolites mechanistically relate to the observed immune and microbiota changes. A more focused narrative around these specific molecular changes, particularly along the taurine/cysteine and tryptophan pathways, would make the conclusions much more compelling.
Finally, although p-values are now indicated for several key outcomes, confidence intervals or measures of effect size are still not reported. Given the relatively small sample size, presenting these would help readers evaluate the robustness of the findings and the biological relevance of the observed differences. This does not require major reanalysis, but simply clearer statistical reporting.
Overall, the manuscript is on a much stronger footing after revision. With these final adjustments, it would meet the level of analytical clarity and interpretive rigor expected for publication.
Author Response
All modifications are highlighted in red.
1.That said, there remain a few key areas that would benefit from further refinement to bring the work fully in line with the standards expected for publication. First, while the inclusion of Spearman correlation analyses provides some insight into the relationship between immune markers and microbial shifts, this approach still falls short of demonstrating any structured interaction between these domains. The manuscript would be significantly stronger with a more explicit statistical or conceptual integration of these datasets, ideally in the form of interaction testing, mediation modeling, or, at the very least, a more rigorous interpretation that goes beyond parallel description.
Response: Added. Line 382-384, Line 386-388.
2.Second, the metabolomics section remains underdeveloped. The authors mention focusing on a few key metabolites, which is a reasonable strategy given the breadth of the data. However, the current discussion still does not clearly articulate how these metabolites mechanistically relate to the observed immune and microbiota changes. A more focused narrative around these specific molecular changes, particularly along the taurine/cysteine and tryptophan pathways, would make the conclusions much more compelling.
3.Finally, although p-values are now indicated for several key outcomes, confidence intervals or measures of effect size are still not reported. Given the relatively small sample size, presenting these would help readers evaluate the robustness of the findings and the biological relevance of the observed differences. This does not require major reanalysis, but simply clearer statistical reporting.
Response: Added to supplementary materials (Appendix 1)
Reviewer 2 Report
Comments and Suggestions for Authors
Dear Authors,
the revised manuscript represents a clear and substantial improvement compared to the initial submission. The manuscript has been notably improved, and the authors have adequately addressed nearly all major concerns raised in the first review. The paper is now well-organized, and the methodological section provides a sufficient level of transparency for reproducibility. Nevertheless, a few points still deserve attention to further strengthen the scientific consistency and readability of the study.
Lines 15-31: Although the abstract is now more concise and within the word limit, the final sentences still sound somewhat assertive “suggest that GOx and MCE can enhance immune function, mitigate oxidative stress…”. It would be preferable to adopt a more cautious tone “may enhance”, “may provide preliminary evidence for”, to better align with the small sample size and exploratory nature of the study.Additionally, mentioning the main limitation (sample size) briefly in the abstract would improve transparency.
Lines 181-193: The revised version now details tests for normality and homogeneity of variance, which is appreciated. However, it remains unclear whether any correction for multiple comparisons was applied in the metabolomic or microbiota datasets. If no correction was used, this should be clearly stated and recognized as a limitation, since some results may include false positives inherent to high-dimensional data.
Lines 329-360: The discussion is now better structured and more mechanistic, but certain sections, particularly those describing cytokines and immunoglobulins, are still dense and could be streamlined. Grouping similar findings and simplifying redundant explanations would enhance readability.
Lines 358-360: The statement that the combined use of GOx and MCE “may exert adverse effects” remains somewhat speculative and should be softened “may indicate a pro-inflammatory trend rather than a clear adverse effect”.
Moreover, although the English writing has improved, minor grammatical and stylistic edits are still advisable. Some metabolite and gene names should be italicized for consistency with the journal’s formatting standards. The reference list now follows the numbered format, but I suggest correcting the remaining minor inconsistencies in journal name abbreviations, so an ISO4 verification would ensure uniformity with MDPI guidelines.
Kind Regards
Author Response
All modifications are highlighted in red.
1.Lines 15-31: Although the abstract is now more concise and within the word limit, the final sentences still sound somewhat assertive “suggest that GOx and MCE can enhance immune function, mitigate oxidative stress…”. It would be preferable to adopt a more cautious tone “may enhance”, “may provide preliminary evidence for”, to better align with the small sample size and exploratory nature of the study. Additionally, mentioning the main limitation (sample size) briefly in the abstract would improve transparency.
Response: Modified. Line 28-33.
2.Lines 181-193: The revised version now details tests for normality and homogeneity of variance, which is appreciated. However, it remains unclear whether any correction for multiple comparisons was applied in the metabolomic or microbiota datasets. If no correction was used, this should be clearly stated and recognized as a limitation, since some results may include false positives inherent to high-dimensional data.
Response: Modified. Line 189-190.
3.Lines 329-360: The discussion is now better structured and more mechanistic, but certain sections, particularly those describing cytokines and immunoglobulins, are still dense and could be streamlined. Grouping similar findings and simplifying redundant explanations would enhance readability.
Response: Modified. Part of the content has been deleted.
4.Lines 358-360: The statement that the combined use of GOx and MCE “may exert adverse effects” remains somewhat speculative and should be softened “may indicate a pro-inflammatory trend rather than a clear adverse effect”.
Response: Modified. Line 362.
5.Moreover, although the English writing has improved, minor grammatical and stylistic edits are still advisable. Some metabolite and gene names should be italicized for consistency with the journal’s formatting standards. The reference list now follows the numbered format, but I suggest correcting the remaining minor inconsistencies in journal name abbreviations, so an ISO4 verification would ensure uniformity with MDPI guidelines.
Response: Modified.